# Quantification of the Dislocation Density, Size, and Volume Fraction of Precipitates in Deep Cryogenically Treated Martensitic Steels

**Ajesh Antony [1], Natalya M. Schmerl [2,\*], Anna Sokolova [3], Reza Mahjoub [2], Daniel Fabijanic [1] and Nikki E. Stanford [2,\*]**

[1] Institute for Frontier Materials, Deakin University, Melbourne, VIC 3216, Australia; ajemetals@gmail.com (A.A.); daniel.fabijanic@deakin.edu.au (D.F.)

[2] Future Industries Institute, University of South Australia, Mawson Lakes Campus, Adelaide, SA 5095, Australia; reza.mahjoub@unisa.edu.au

[3] ACNS, ANSTO, New Illawarra Rd, Lucas Heights, NSW 2234, Australia; anna.sokolova@ansto.gov.au

\* Correspondence: natalya.schmerl@unisa.edu.au (N.M.S.); nikki.stanford@unisa.edu.au (N.E.S.)

**Abstract:** Two groups of martensitic alloys were examined for changes induced by deep cryogenic treatment (DCT). The first group was a range of binary and ternary compositions with 0.6 wt % carbon, and the second group was a commercial AISI D2 tool steel. X-ray diffraction showed that DCT made two changes to the microstructure: retained austenite was transformed to martensite, and the dislocation density of the martensite was increased. This increase in dislocation density was consistent for all alloys, including those that did not undergo phase transformation during DCT. It is suggested that the increase in dislocation density may be caused by local differences in thermal expansion within the heterogeneous martensitic structure. Then, samples were tempered, and the cementite size distribution was examined using small angle neutron scattering (SANS) and atom probe tomography. First principles calculations confirmed that all magnetic scattering originated in cementite and not carbon clusters. Quantitative SANS analysis showed a measurable change in cementite size distribution for all alloys as a result of prior DCT. It is proposed that the increase in dislocation density that results from DCT modifies the cementite precipitation through enhanced diffusion rates and increased cementite nucleation sites.

**Keywords:** martensite; tempering; cryogenic treatment; small angle scattering; atom probe tomography; X-ray diffraction

## 1. Introduction

Deep cryogenic treatment (DCT) is a sub-zero treatment where materials are exposed to temperatures below −125 °C with liquid nitrogen being the preferred cooling media [1]. Deep cryogenic treatment on martensitic steels is commercially carried out to improve wear resistance [2–7], and it has also been reported to affect other material properties such as hardness [8–20] and tensile strength [21–23]. There are some proposed mechanisms for these improvements in the literature. The transformation of retained austenite (RA) to martensite is one of the primary objectives of DCT, and it is one of the predominating proposed mechanisms for improvement in wear resistance, as the martensitic phase is much harder than the parent austenite phase [2,4,19]. This transformation has regularly been quantified and verified via X-ray diffraction (XRD) [3,4,19,22,24–30]. However, it has also been observed that alloys that are already fully martensitic can exhibit markedly improved wear resistance after DCT, even though there is no change to the martensite volume fraction [31]; this is commonly referred to as the conditioning of the martensite [32].

Industrially, martensitic tool steels are tempered before use. When DCT is applied, the usual process is to quench (or cool) the alloy to room temperature and then carry out the DCT, which is followed by a tempering treatment. The most commonly proposed mechanism for improvement in wear resistance other than the removal of RA after DCT is an increase in the volume fraction and homogeneity of secondary alloy carbides that develop during tempering [4,19,21,24,30,32–34] or during the DCT itself [22]. Some work on tool steels has claimed that carbon atoms segregate to dislocations during the DCT [20,23,30,33,35–37], and that it is these segregated atoms that contribute to the increase in carbides during the tempering process [28]. However, other arguments state that the carbon atoms are essentially immobile at the temperatures used in DCT [38,39] and that it is in fact gliding dislocations that capture immobile carbon atoms [4,40], and some work has reported an elimination of fine carbide precipitation as a result of DCT [41]. Therefore, it remains unclear what effect DCT has on the subsequent tempering behavior and carbide development, with a range of results being published [2,39,42]. This is further complicated by the observation that DCT affects the properties of steels with varying carbon concentrations differently [43,44]. The effect of DCT on carbide precipitation remains an open question.

Another hypothesis for the improvement in wear properties resulting from DCT is an increase in dislocation density in the martensite. However, despite this being repeatedly referred to in the literature [2,10,18,20,27,34,38,39,45], there is minimal quantitative analysis of dislocation density available. The internal friction method has been used to quantify dislocation density [37,46,47], but this is not a conventional technique for this kind of measurement. Zare et al. [22], Li et al. [3] and Li et al. [48] have examined changes to the dislocation density using transmission electron microscopy, but this only measures the dislocation density within a very small volume of an extremely inhomogeneous microstructure. Abudaia and Fessatwi [31] carried out an XRD investigation into the effect of DCT on a martensitic steel, and they noted peak broadening in the diffractograms after DCT but failed to carry out a quantitative dislocation density analysis. Thus, the effect of DCT on the dislocation density of martensitic alloys remains an open question that we address here with quantitative XRD measurements.

Another open question in the literature is the effect of soaking time during DCT. It has been reported that soaking for longer times can improve the wear resistance of martensitic steels [22,33,39]. Conventional thought on the martensitic transformation is that the austenite will transform to martensite almost immediately upon reaching the required temperature, and additional time at this temperature will not change the volume fraction of martensite. It also seems unlikely that significant changes could occur during a soak at such low temperatures due to the much reduced rates of diffusion at cryogenic temperatures [38]. In order to clarify if there is an effect of soak time, several times are tested here, which are based on reports in the literature.

## 2. Materials and Methods

### 2.1. Alloy Design

Alloy design was centered on the Schaeffler diagram and aimed at producing microstructures with varying retained austenite through the addition of Ni or Cr. Utilizing both Ni and Cr allows assessing the effect of composition on the chemistry of the precipitates formed during DCT and tempering. Specifically, prior studies [7,9] have suggested that chromium-rich alloy carbides can form after 180–200 °C tempering of DCT-treated martensite.

Two groups of alloys were produced: a set of model alloys and a set of commercial alloys; see Tables 1 and 2. The model alloys for this work were made in-house in 20 kg batches using an induction furnace to melt and cast the alloys. These model alloys have binary and ternary compositions but all with a nominal carbon concentration of 0.6 wt %, and can be summarized as follows:

- Binary Fe C alloy: Fe—0.6C
- Ternary Fe-C-Ni alloy: 2, 4, 12 and 20 wt % Ni
- Ternary Fe-C-Cr alloy: 4, 7 and 11 wt % Cr

**Table 1.** Chemical Composition Determined Using a SPECTROMAXx Stationary Metal Analyzer (SPECTRO Analytical Instruments GmbH, Kleve, Germany) in Weight Percent of the Cast Model Alloys.

| Type | Alloy | Composition (wt %) | | | |
|---|---|---|---|---|---|
| | | Fe | C | Ni | Cr |
| Base Alloy | Fe-0.6C | 99.2 | 0.57 | <0.0015 | 0.009 |
| Ni | Fe-0.6C-2Ni | 97.1 | 0.55 | 2.1 | 0.019 |
| | Fe-0.6C-4Ni | 95.2 | 0.63 | 3.9 | 0.019 |
| | Fe-0.6C-12Ni | 87.2 | 0.66 | 11.8 | 0.021 |
| | Fe-0.6C-20Ni | 79.4 | 0.55 | 19.7 | 0.018 |
| Cr | Fe-0.6C-4Cr | 95.2 | 0.54 | 0.014 | 4.1 |
| | Fe-0.6C-7Cr | 91.7 | 0.57 | 0.017 | 7.5 |
| | Fe-0.6C-11Cr | 88.8 | 0.53 | 0.016 | 10.4 |

**Table 2.** Chemical Composition Determined Using a SPECTROMAXx Stationary Metal Analyzer in Weight Percent of the AISI D2.

| Element | Composition (wt %) | Element | Composition (wt %) |
|---|---|---|---|
| C | 1.568 | Cu | 0.085 |
| Si | 0.275 | Nb | 0.017 |
| Mn | 0.266 | Ti | 0.002 |
| P | 0.007 | V | 0.69 |
| S | <0.0005 | W | 0.114 |
| Cr | 12.26 | Zr | 0.01 |
| Mo | 0.694 | B | 0.0005 |
| Ni | 0.23 | Zn | 0.0006 |
| Al | 0.031 | N | 0.091 |
| Co | 0.054 | Fe | 83.58 |

The second group of samples contains all commercially sourced AISI D2 tool steel (Bohler Uddenholm), and within this group, the only variable was the austenitizing temperature. Again, the motivation to explore varying austenitizing temperatures of the AISI D2 was to induce differing quantities of retained austenite. Higher austenitizing temperatures result in an increased dissolution of secondary and primary carbides. The resulting elevated carbon content in the matrix assists to stabilize the austenite phase upon quenching.

*2.2. Heat Treatments*

The nickel-containing alloys were austenitized at 1000 °C in a fluid bed furnace, and the chromium-containing alloys were austenitized at 1150 °C in a muffle furnace. The higher temperature was required for the Cr-containing alloys to dissolve primary carbide phases formed during solidification. Austenitizing for all model alloys was carried out for 30 min in an inert atmosphere followed by an immediate water quench. The austenitizing of AISI D2 steel was performed at various temperatures; 980, 1020, 1040, 1060, and 1080 °C, holding the sample in an inert atmosphere at temperature for 30 min before water quenching. All tempering was carried out in a muffle furnace, in air, at 200 °C for 2 h.

*2.3. Cryogenic Treatments*

A cryogenic treatment chamber manufactured by Cryotron© (Cryotron (Canada) Ltd., Spruce Grove, Canada) was used in this project. Cooling to the cryogenic treatment chamber was supplied by liquid nitrogen, which was delivered by a temperature controller. Samples were cooled to −196 °C at 1 °C per minute and then soaked for predominately 24 h, although soaking times of 4 and 36 h were

also applied to select alloys. The chamber was switched off post treatment so that natural heating was used to allow samples to equilibrate to room temperature; this process took 400 min.

### 2.4. Optical and Electron Microscopy

Standard metallographic sample preparation techniques were used to prepare samples for optical microscopy (Olympus SC50, Tokyo, Japan), XRD (details in Section 2.6), and scanning electron microscopy (SEM) described below. Samples were mechanically ground using 80-grit SiC paper and finished with 1200-grit SiC paper. This was followed by polishing with a 15 μm to 1 μm diamond paste. Final polishing was carried out using colloidal silica for at least 10 min to remove the deformed surface layer. SEM was performed at 25 kV on a Zeiss Supra 55VP (Jena, Germany) equipped with a field emission electron source. For optical microscopy, polished samples were etched in a mixture of 5% nitric acid in ethanol, which is commonly referred to as nital.

### 2.5. Small Angle Neutron Scattering

#### 2.5.1. Instrumentation

Small angle neutron scattering (SANS) experiments were conducted using the instrument known as Bilby [49] located at the Australian Center for Neutron Scattering (ACNS), Australian Nuclear Science and Technology Organization (ANSTO). Bilby is a time-of-flight mode instrument that operates over a wavelength range from 2 to 20 Å. Six detector panels are located inside a large vacuum chamber to detect the neutrons. All measurements were made at room temperature in a magnetic field of 1 Tesla, which has been shown to be sufficient to magnetically saturate the ferritic matrix [50]. Data obtained from wavelengths between 4.2 and 10 Å were chosen for reduction in this study because below 4.2 Å, a Bragg's edge was observed, and above 10 Å, multiple scattering was observed (details of the Braggs edge and multiple scattering events are given in Appendices A and B). Multiple scattering in reduced data generates incorrect analysis values, altering data at low scattering angles [49].

#### 2.5.2. Small Angle Neutron Scattering from Cementite

In steels that have a large volume fraction of precipitates with a large chemistry difference to the matrix, the nuclear scattering signal is sufficiently large for quantitative analysis [51]. However, in most steels, the volume fraction of precipitates is small, and magnetic scattering is utilized to amplify the signal [52,53]. In the present case, the samples were examined in a constant magnetic field of 1 Tesla, which is a value chosen to ensure that the magnetic matrix was magnetically saturated. Any precipitate that is non-magnetic will produce a "hole" in the magnetic matrix and give rise to magnetic scattering. The magnetic properties of the different phases in the material determines if they will give rise to magnetic scattering. Martensite is ferromagnetic, but cementite is also ferromagnetic [54]. However, it is pointed out here that the magnetic scattering which is measured in the neutron scattering experiment is proportional to the magnetic susceptibility of the material being studied. The detailed mathematical argument for this has previously been described in the literature [55,56], and it is available here in Appendix C. The magnetic susceptibility of iron is $2 \times 10^5$ [57] and it is by far larger than cementite, which is estimated to be around 2.6 [58]. Thus, the cementite, despite being ferromagnetic, provides negligible magnetic scattering in the present experiment, and we therefore can consider its magnetic scattering length density to be zero.

#### 2.5.3. Small Angle Neutron Scattering from Clusters—First Principles Calculations

In addition to discrete cementite particles, martensite is known to form small carbon-enriched clusters and nano-precipitates [59]. The ability of these regions to produce magnetic scattering during SANS has not been examined before. It has been reported that the experimentally measured magnetic susceptibility of BCC (body centred cubic) iron decreases as the carbon content is increased [60]. As a consequence, the magnetic saturation of the BCC matrix may be weakened in localized regions of

high carbon concentration, such as solute clusters. These may or may not provide a magnetic scattering signal during SANS. Therefore, first principles calculations are used here to determine the effect of interstitial carbon concentration on magnetic susceptibility, noting again that these regions will likely be ferromagnetic, and their susceptibility will determine their scattering capacity.

To determine the effect of interstitial carbon concentration on magnetic susceptibility, ab initio spin-polarized calculations based on density functional theory (DFT) were performed to calculate the magnetic moment per atom of a series of interstitial C-doped Fe alloys. The calculation program known as VASP was used [61] implementing the projector augmented wave method to represent the combined potential of core electrons and nuclei [62]. The Perdew–Burke–Ernzerhof gradient approximation is implemented to represent the exchange–correlation functional [63]. A cut-off energy of 400 eV was chosen for the plane wave basis, and the self-consistent electronic optimization was converged to $10^{-7}$ eV. The supercell contained 54 atoms with BCC symmetry and dimensions of $3a \times 3a \times 3a$ (a = 2.866 Å). Periodic boundary conditions were applied in three dimensions and a mesh of $3 \times 3 \times 3$ Γ-centered k-points were chosen to sample the Brillouin zone.

### 2.5.4. Quantitative Fitting Procedure

Quantitative fitting of the SANS data was carried out using the commercially available software package IGOR Pro, using the NIST (national institute of standards and technology) small angle scattering macro written by Kline [64]. Quantitative fitting was only carried out on the magnetic signal and assumed that all scattering species were spherical and non-magnetic [65]. The magnetic scattering length density of the martensitic matrix, $\rho_{mag}{}^{\alpha}$, is $5 \times 10^{-6}$ Å$^{-2}$ and the non-magnetic scattering species, $\rho_{mag}{}^{\beta}$, is zero [50], and $\Delta\rho_{mag} = \rho_{mag}{}^{\alpha} - \rho_{mag}{}^{\beta} = \rho_{mag}{}^{\alpha}$.

The particles were assumed to have a Schulz size distribution:

$$f(R) = (z+1)^{z+1} x^z \frac{exp[-(z+1)x]}{R_{avg}\Gamma(z+1)} \tag{1}$$

where

$\sigma$ = the standard deviation
$R$ = radius
$R_{avg}$ = mean radius
$x = \frac{R}{R_{avg}}$
Polydispersity, $p = \frac{\sigma}{R_{avg}}$
$z = \frac{1}{p^2} - 1$.
With a Schulz distribution, the intensity $I_{mag}(q)$ is calculated by:

$$I_{mag}(q) = \left(\frac{4\pi}{3}\right)^2 N_0 \Delta\rho_{mag}{}^2 \int_0^\infty f(R)R^6 F^2(qR)dR \tag{2}$$

where $N_0$ is the total number of particles per unit volume, $f(R)$ is the particle size distribution, and $F(qR)$ is the scattering amplitude for a sphere, which is calculated as:

$$F(x) = \frac{3[\sin(qR) - x\cos(qR)]}{qR^3}. \tag{3}$$

To fit the SANS data using the Schulz distribution, the polydispersity of the small and medium-sized particles was constrained to be close to 0.5.

### 2.6. X-ray Diffraction

X-ray diffraction (XRD) was used to quantify two different parameters: the volume fraction of retained austenite and the dislocation density of the martensite. X-ray measurements were carried

out with line focus optics in either a PANalytical materials research diffractometer (MRD) using Cu Kα radiation and a Ni filter, or a Bruker D8 ECO with Co Kα radiation and an Fe filter. Instrumental broadening was characterized using a NIST standard Lanthanum hexaboride (LaB$_6$) powder placed on a silicon single crystal wafer. The micro-strain ($\varepsilon_o$) as determined by peak width and phase data were fitted to the spectrum using the software package TOPAS 5.0. For phase determination, all martensite was assumed to be a BCC structure. Quantitative phases analysis was carried out using TOPAS 5.0 using the full diffractogram. The reflection selected for evaluation of the micro-strain was between 80 and 85° 2θ angle with a step size of 0.02° at various psi values between 0 and 50°. It was assumed that grain size broadening was negligible, and that the only source of micro-strain in the sample was from the dislocations. Thus, micro-strain was used to quantify the dislocation density using the following relationship and values for the burgers vector and $k$ [66]:

$$\rho = \frac{k\varepsilon_0^2}{b^2} \tag{4}$$

$\rho$ = dislocation density (m/m$^3$)
$\varepsilon_0$ = micro-strain
$b$ = Burgers vector = 0.248 nm
$k$ = 14.4.

### 2.7. Atom Probe Tomography

For atom probe tomography (APT), specimens were first electro-polished using a standard micro-loop apparatus and a 5% nitric acid in ethanol solution under an optical microscope. Then, the pre-polished tips were given a final sharpening treatment with a focused ion beam milling machine using an FIE Quanta FIB SEM. Atom probe tomography data were obtained using a LEAP 4000X HR in a voltage pulse mode at 200 kHz with a pulse fraction of 20% and a detection rate of 1.0%. Then, the specimen was cryogenically cooled to 50K and visually aligned pointing toward the local electrode. The reconstruction, mass ranging, and data analysis was carried out using the IVAS software.

## 3. Results Part 1: Effect of Deep Cryogenic Treatment

### 3.1. Microstructure

The starting condition for this study was solution treatment in the austenite phase field followed by an immediate water quench. In this starting condition, the alloys were either fully martensitic or they were a mixture of martensite and retained austenite; see Figure 1. The volume fraction of retained austenite was quantified using XRD; see Figure 2 (XRD diffractograms are provided in Appendices D and E). This analysis showed that the alloys with a low Ni concentration were largely martensite, but the highest Ni alloy was fully austenitic at room temperature. The D2 tool steel retained between 10 and 20% austenite, with the higher solution treatment temperatures stabilizing austenite through carbide dissolution during the solution treatment cycle. Although not shown here in the interests of brevity, the volume fraction of Cr$_7$C$_3$ was also quantified and found to drop from 20% at the lowest solution treatment temperature to 15% at the highest solution treatment temperature. In those alloys with substantial retained austenite volume fractions, cryogenic treatment resulted in the substantial transformation of austenite to martensite.

The Vickers hardness of all samples were measured in the water quenched and cryogenically treated conditions. In alloys that exhibited large transformation volumes due to cryogenic treatment, a large increase in hardness was observed; see Figure 3. This corresponds to the softer austenite phase being transformed to the harder martensite phase. It is noteworthy that for those alloys that began in the fully martensitic condition, such as the Fe-0.6C alloy, an increase in hardness was also observed after DCT. In these alloys, there was no change in constituent microstructure, as they were both fully martensitic in the water-quenched condition, yet cryogenic treatment induced a hardening

effect. This observation was further interrogated using XRD to measure the dislocation density of the martensite; see Figure 4. Here, it can be seen that there is a consistent increase in dislocation density of the martensite in all alloys after cryogenic treatment. Cryogenic treatment time shows no effect on the measured values of dislocation density or hardness.

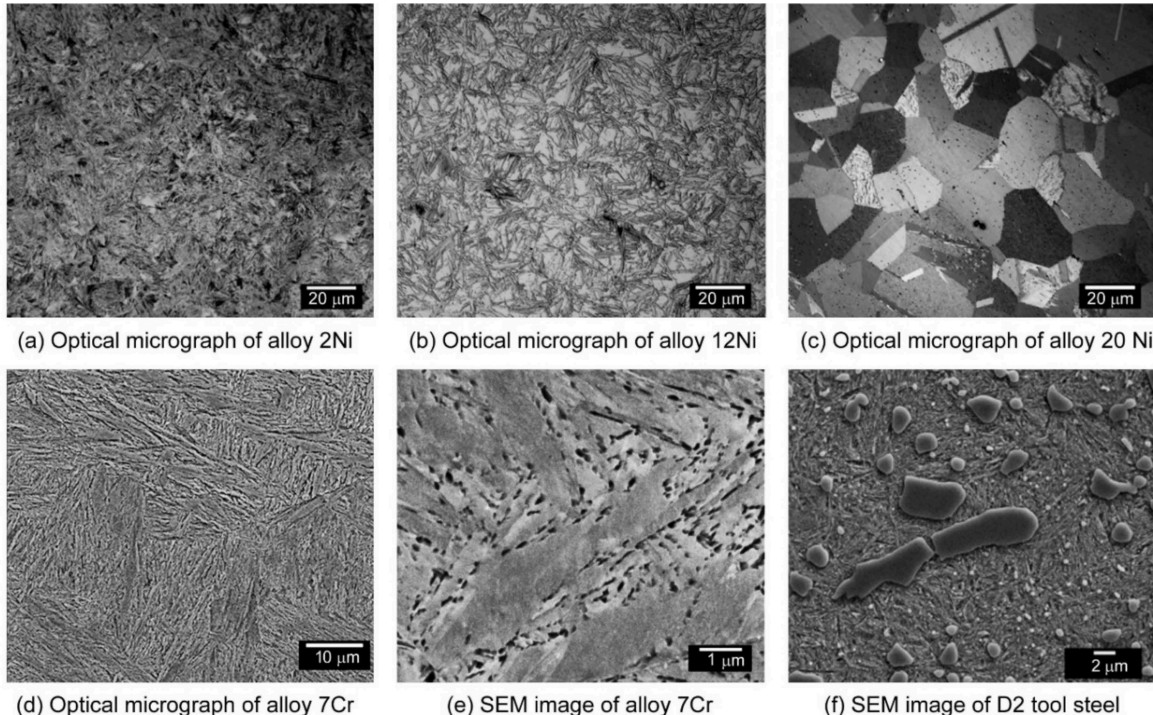

**Figure 1.** Microstructures of selected alloys from the present study. (**a**) Optical micrograph of martensitic 2Ni alloy. (**b**) Optical micrographs of mixed austenite and martensite microstructure in 12Ni alloy. (**c**) Optical micrograph of austenitic 20Ni alloy; note that the small black dots are an etching artefact. (**d**,**e**) Martensitic structure of 7Cr alloy, black dots in (**e**) result from etching. (**f**) Carbides and martensite in D2 tool steel. Note higher magnification in (**e**,**f**).

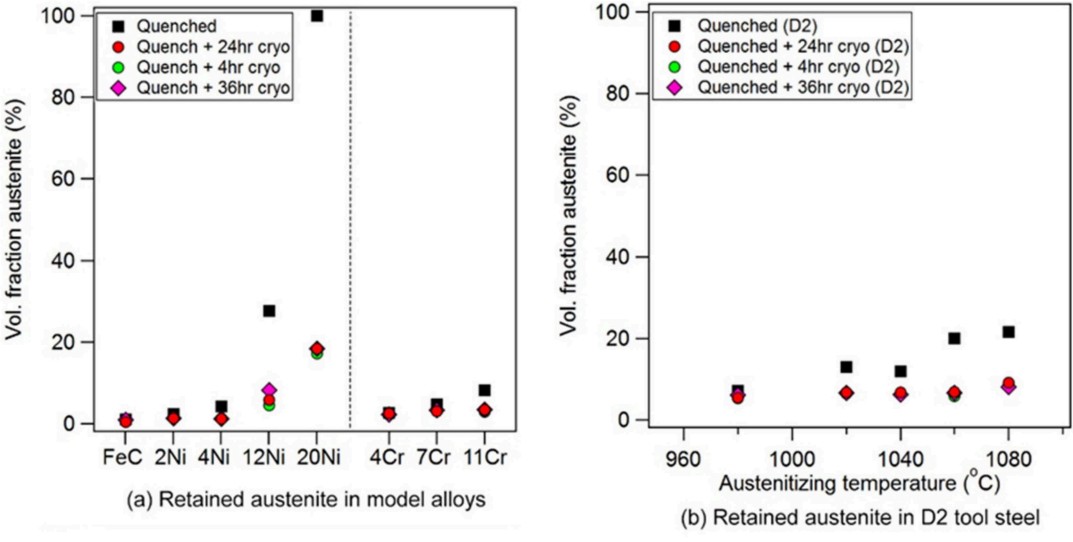

**Figure 2.** Volume fraction of retained austenite for all alloys examined in this study, which were measured using XRD. Data shown for cryogenic soaking times of 4, 24, and 36 h. The fitting error is approximately the same size as data points.

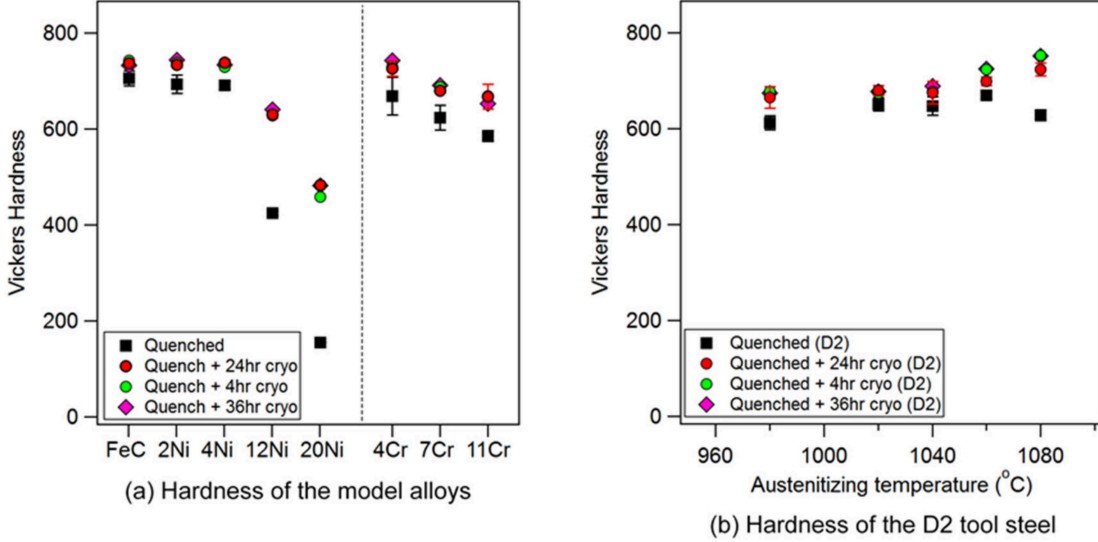

**Figure 3.** Vickers hardness of martensite for all alloys examined in this study. Data shown for cryogenic soaking times of 4, 24, and 36 h. Fitting error approximately the same size as data points.

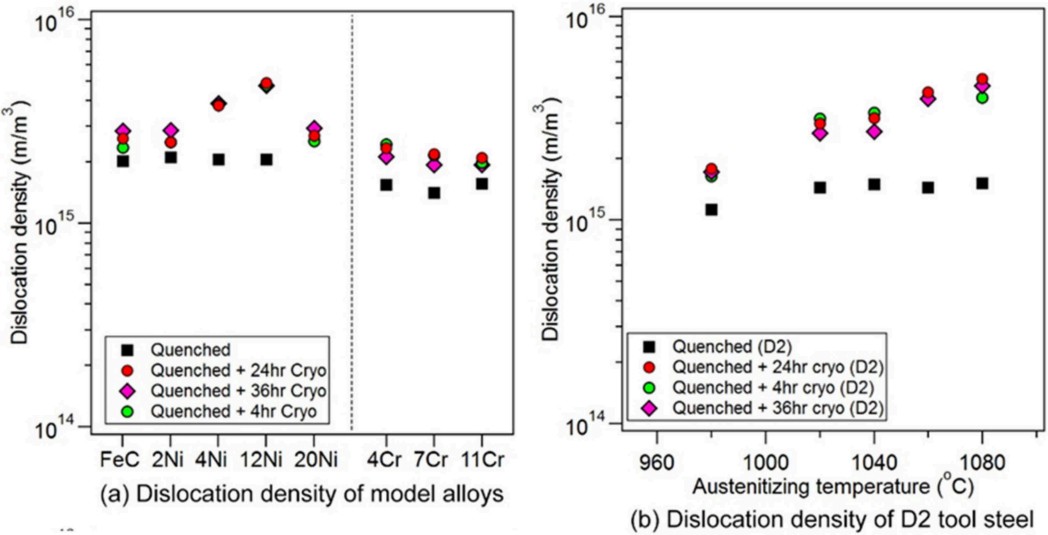

**Figure 4.** Dislocation density of martensite for all alloys examined in this study, measured using XRD. Data shown for cryogenic soaking times of 4, 24, and 36 h. Fitting error approximately the same size as data points.

### 3.2. Small Angle Neutron Scattering

A typical scattering signal obtained in this study is shown in Figure 5a. The pattern shows the classical apple shape that results from the additional scattering intensity perpendicular to the applied magnetic field. To separate the nuclear and magnetic signals from the two-dimensional detector panel, sections of data from a 15° arc parallel to the nuclear and the nuclear + magnetic directions were segmented for further processing. The gap on the SANS data presented in this paper is caused by a miss in the experimental setup. The experiment described here was the first of its kind on ToF (time of flight) reactor-based SANS. Care has been taken to ensure production of the continuous datasets, but the amplitude of the multiple scattering issue has been overlooked. During data reduction, we had to truncate the substantial amount of long wavelengths, losing some data. The issue is avoidable by selecting different configuration of detectors and making the samples thinner.

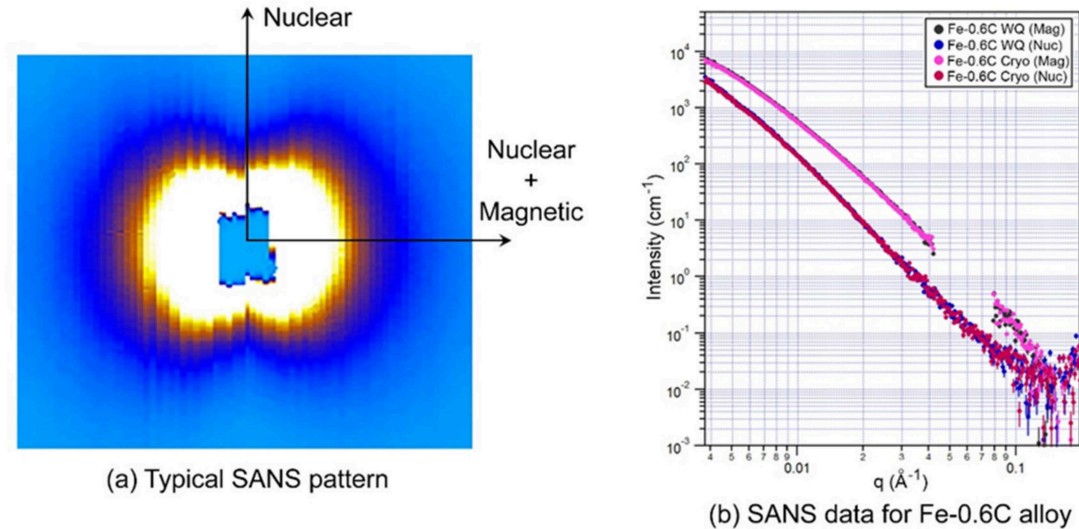

**Figure 5.** (**a**) Typical scattering pattern obtained from the martensitic alloys. The scattering pattern was measured from a sample during exposure to a 1T magnetic field. (**b**) Nuclear and magnetic scattering signal from martensitic Fe-0.6C alloy in two conditions: water quenched; and water quenched followed by 24 h deep cryogenic treatment.

The SANS data for the Fe-0.6C alloy in the as-quenched and cryogenically treated conditions are shown in Figure 5b. It can be seen there is no difference between the scattering signals for the two different conditions tested; the scans were identical. It can be concluded that DCT has no effect on those microstructural features that are detected by SANS, specifically, solute distribution in features such as particles and clusters.

Since no difference was detected in this first experiment, and because there was limited beam time, no further SANS measurements were carried out on samples in the quenched, and cryogenically treated conditions. All further SANS measurements were made on the tempered specimens. These are described in Section 4.2.

It is noted here that the atom probe tomography of selected specimens was carried out, but no conclusion about the effect of DCT on the microstructure or solute distribution could be made due to the inherently complex nature of martensitic microstructures, which give rise to large variations in the local composition and structure [59].

### 3.3. Atom Probe Tomography

Atom probe tomography was also carried out on quenched and cryogenically treated specimens. This revealed that, consistent with the literature, there exists a wide variety of carbon distributions within the same sample, which depends on the location of sampling; see Figure 6. Due to this inherent variability in local solute distribution, no conclusions could be made about the effect of DCT on microstructure using atom probe tomography. There was significant variation between volumes even within the same specimen, and this is consistent with the recent characterization of martensitic structures [59]. Only one volume is shown here as an example of the data obtained. The carbon was visibly non-random in all atom probe experiments (see Figure 6). Clusters finding was carried out but has not been shown, as the results provided no further insight to this work; the results were similar to those shown in [59]. As such, the claims made by Xie et al. [26,46] were not able to be verified or expanded upon in this experiment.

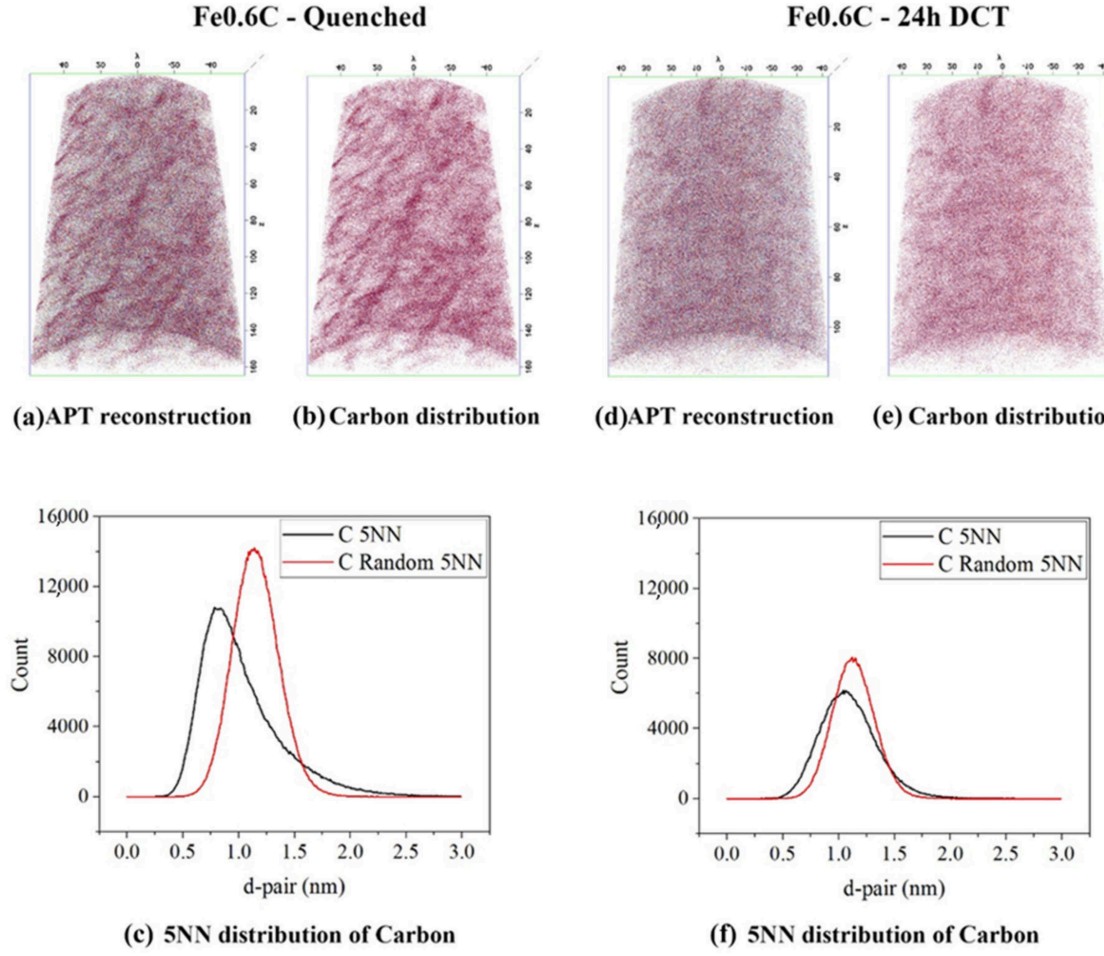

**Figure 6.** Atom probe tomography of Fe-0.6C alloy. All volumes were measured from atom probe tips produced from the same sample and show the inherent variability in solute distribution within martensitic structures. 5NN represents that the five nearest neighbors have been used for the data analysis.

## 4. Results Part 2: Effect of Tempering

### 4.1. Tempered Microstructure

Tempering decreased the hardness of all samples and also corresponded to a decrease in dislocation density; see Figures 7 and 8. Atom probe tomography was carried out on selected samples and in the interest of brevity, only one example is shown here. The Fe11CR sample shown here was of particular interest because in the literature [7,9,46], it had been suggested that after cryogenic treatment, Cr carbides are redistributed throughout the microstructure, thereby enhancing the wear properties. One of the atom probe volumes obtained is shown in Figure 9. The carbon map clearly shows segregated regions. Since the equilibrium precipitate is $Fe_3C$, an iso-concentration of 12.5 at.% was chosen to highlight clusters of significance. It can be seen that there are two populations of carbon-rich clusters: one large precipitate >100 nm in length, and several small clusters <50 nm in length. A proxigram was created from the largest particle in the volume, and it was seen that Cr is not partitioned into this precipitate. This is also evident in a qualitative way in the Cr map in Figure 9. Thus, it can be concluded that in this sample, the precipitates that form during tempering are Fe-carbide in nature, they do not contain Cr, and they have a bimodal particle size distribution. Many more APT volumes were examined, too many to show here, and it was found that all alloys produce iron–carbide clusters, and that the larger ones had a composition close to $Fe_3C$.

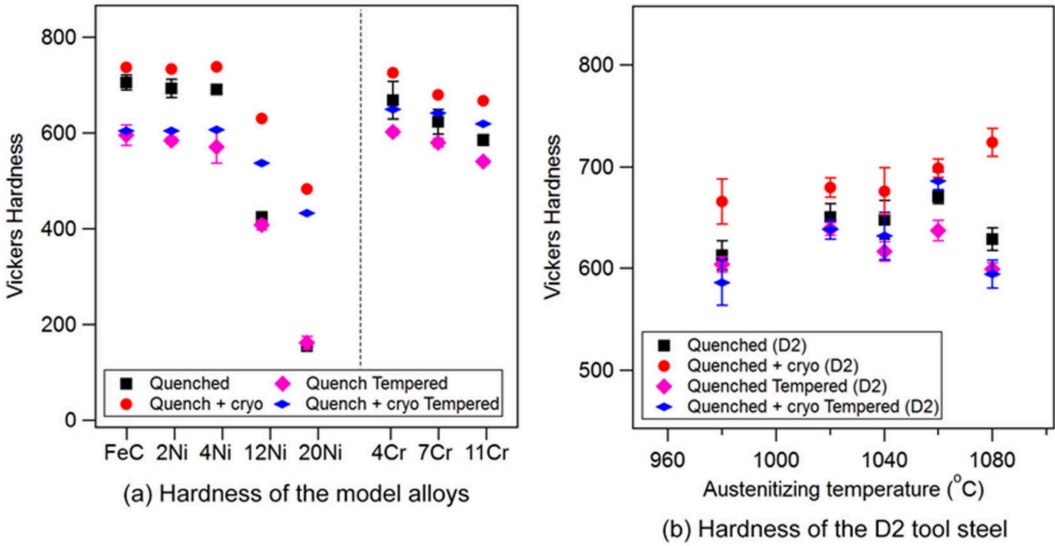

**Figure 7.** Vickers hardness of samples before and after tempering treatment.

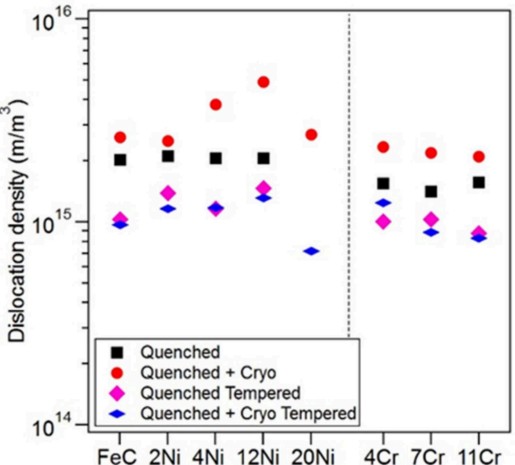

**Figure 8.** Dislocation density of the model alloys before and after tempering, which was measured using XRD. The cryogenic treatment for these specimens was 24 h. The fitting error is approximately the same size as the data points.

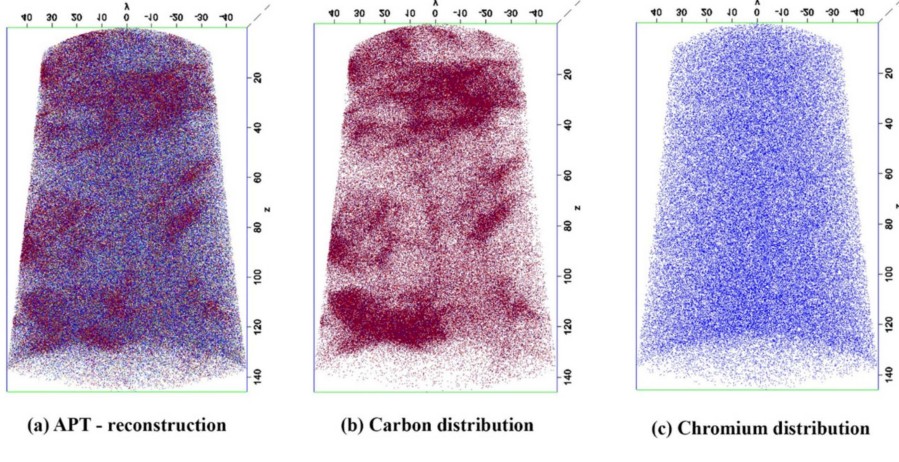

**Figure 9.** *Cont.*

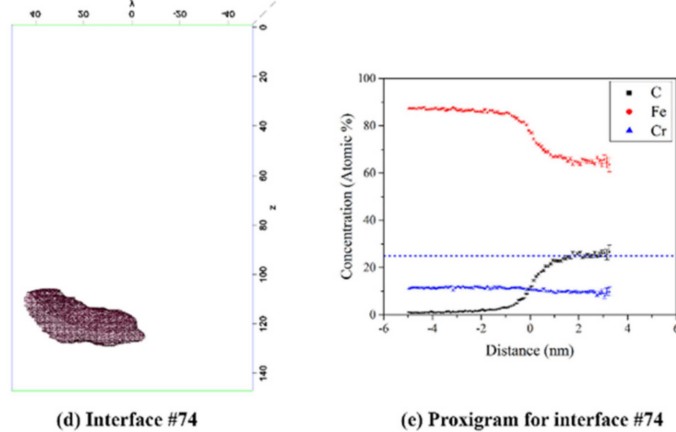

**(d) Interface #74**        **(e) Proxigram for interface #74**

**Figure 9.** Example of the atom probe data obtained from tempered martensite microstructures. Alloy shown is 11Cr alloy after deep cryogenic treatment (DCT) and tempering. Dotted line in (**e**) represents stoichiometric $Fe_3C$, 25 at.% C. Note the lack of Cr partitioning to any feature within the analyzed volume.

### 4.2. Small Angle Neutron Scattering of Tempered Samples

SANS was used to quantify the effect of cryogenic treatment on the tempered microstructure. It was seen that in contrast to the quenched condition, the tempered samples did show some differences depending on whether or not they were cryogenically treated prior to tempering; see Figure 10a. Some samples showed two distinct peaks in the scattering signal, such as Figure 10b. As a result of this observation, the fitting was carried out using two particle populations for all alloys. These populations will hereafter be referred to as the fine particles (radii from 3–10 nm) and the medium-size particles (around 30 nm average radius).

### 4.2.1. First Principles Calculations of Magnetic Properties

First principles calculations were carried out to examine the effect of interstitial carbon concentration in the BCC matrix on the magnetic properties of the local volume. This was performed to determine if the solute clusters within the BCC matrix have the capacity to scatter, or if we only obtain scattering signal from cementite particles. The results of this analysis are shown in Figure 11. It is seen that the increase in carbon concentration leads to the decrease of magnetic moment M(r) and therefore a decrease in magnetic susceptibility χ (since M(r)∝χ). However, extensive atom probe tomography studies of martensite published in the literature show that solute clusters of carbon in martensite are not typically above ≈8 at.% carbon [59,67]. At a local concentration of 8 at.% carbon, Figure 11 shows that these regions would have ≈80% of the magnetization compared to the surrounding matrix. Therefore, any scattering signal from them would be very weak. Therefore, it can be concluded that the solute clusters found in martensite are unlikely to give a significant magnetic signal from SANS; thus, all scattering species in the present case are cementite.

### 4.2.2. Quantitative Fitting of Small Angle Neutron Scattering Data for Tempered Samples

The outcome of the quantitative fitting procedure is shown in Figure 12 where the particle size distributions are plotted as a function of volume fraction. Four alloys had compositions such that they had only small (or negligible) volume fractions of retained austenite; thus, cryogenic treatment had minimal changes on the martensite fraction. In these cases, FeC, 4Ni, 11Cr, and D2, the effect of cryogenic treatment on the tempered particle distributions are directly comparable. It was seen that for three of these martensitic alloys, FeC, 4Ni, and D2, cryogenic treatment decreases the overall volume fraction of particles in the medium-size range. In the case of the 11Cr sample, the volume fraction of the medium-sized particles was similar in the two conditions, and the average particle size was decreased. For the case of the Fe12Ni sample, there is an appreciable difference in the volume fraction

of martensite between the quenched and cryogenically treated sample. Thus, in this case, the volume fractions cannot be directly correlated. However, it was seen that similar to the 11Cr, the average size of the medium-sized precipitates was reduced by the cryogenic treatment. For the 20Ni sample, only the cryogenically treated sample was examined, because no martensite was present in the quenched condition. It is also interesting to note the difference between the different martensite compositions. As the alloying content increases, the relative fraction of the medium size particles as compared to the small particles increases quite markedly (for example, compare Figure 12a,f).

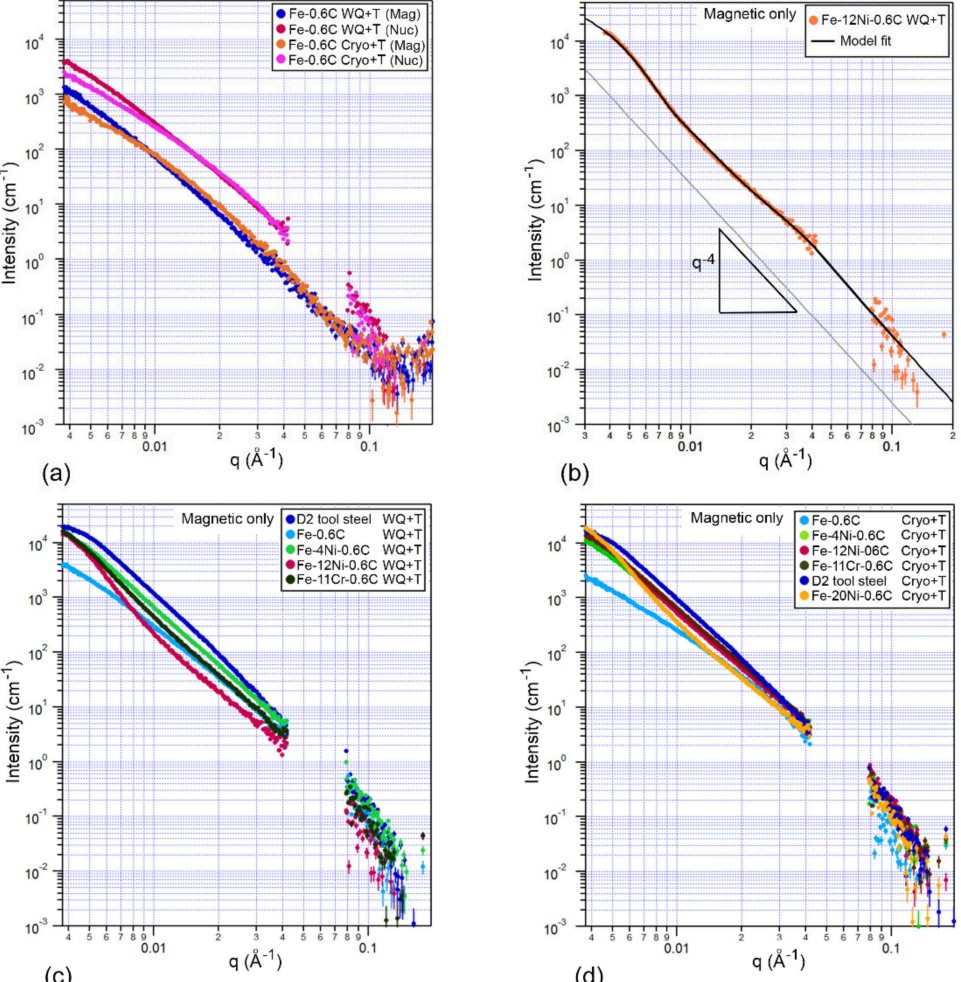

**Figure 10.** Examples of small angle neutron scattering (SANS) patterns measured on various samples after tempering. Alloys were subject to two different treatments before tempering: water quenching (WQ + T); and water quenching followed by deep cryogenic treatment (Cryo + T). (**a**): Fe-0.6C alloy: (blue and red), magnetic and nuclear scattering, respectively, water quenched before tempering; (orange and purple), magnetic and nuclear scattering, respectively, water quenched followed by deep cryogenic treatment before tempering. (**b**): (orange): Magnetic SANS data from Fe-0.6C-12Ni water quenched before tempering alloy; (solid black) calculated pattern of $q^{-4}$, a modeled expected asymptote for intensity scattered by a precipitate separated from the matrix by a sharp interface has an asymptotic behavior described by the Porod law. (**c**): Magnetic SANS data from five alloys, water quenched before tempering: (blue) D2 tool steel; (light blue) Fe-0.6C; (green) Fe-0.6C-4Ni; (red) Fe-0.6C-12Ni; (black) Fe-0.6C-11Cr. (**d**): Magnetic SANS data from six alloys, water quenched followed by deep cryogenic treatment before tempering: (light blue) Fe-0.6C; (green) Fe-0.6C-4Ni; (red) Fe-0.6C-12Ni; (black) Fe-0.6C-11Cr; (blue) D2 tool steel; (orange) Fe-0.6C-20Ni.

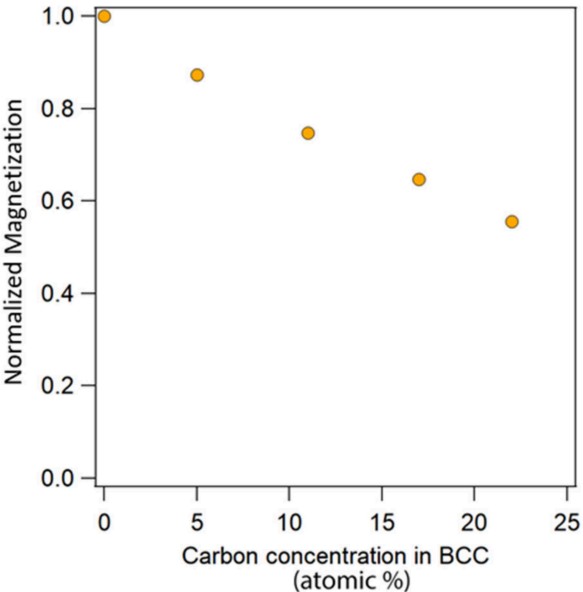

**Figure 11.** The normalized magnetization of BCC iron with interstitial carbon.

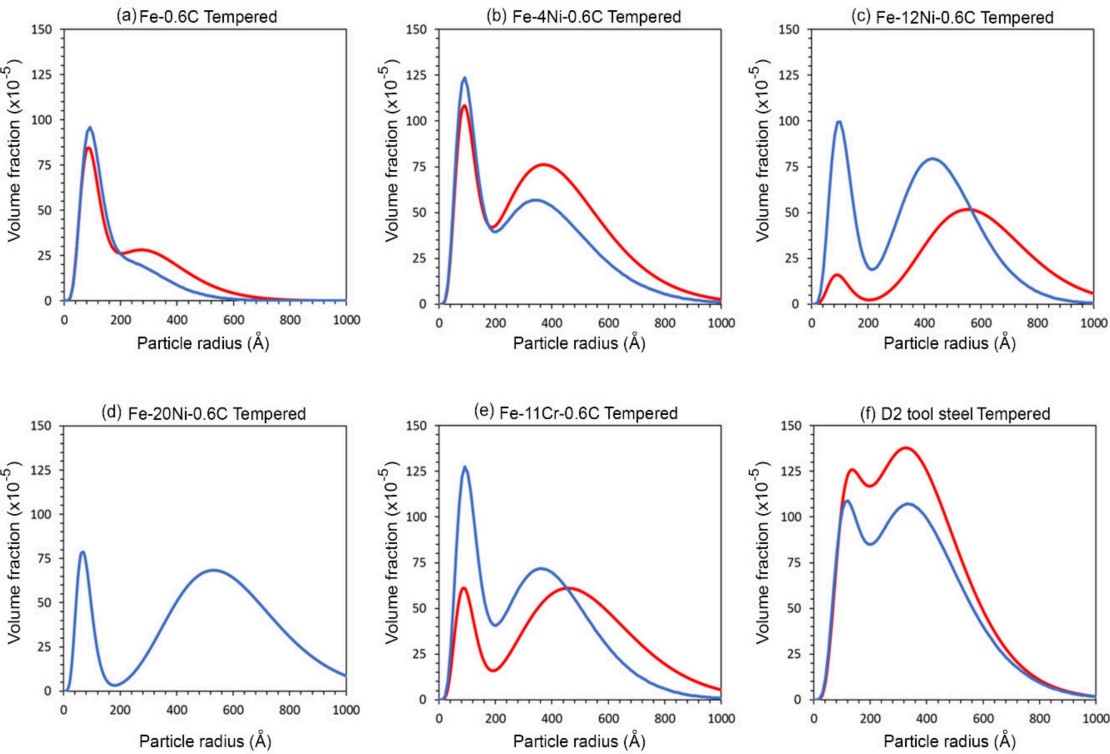

**Figure 12.** Particle size distribution resulting from tempering. Measured using quantitative fitting of the SANS magnetic signal. The red line indicates the Schulz distribution of non-magnetic particles in conventionally treated alloys, and the blue line indicates the Schulz distribution of non-magnetic particles in cryogenically treated alloys.

## 5. Discussion

### 5.1. Effect of DCT on Microstructure

In alloys that had retained austenite in the microstructure after water quenching, DCT resulted in a transformation of retained austenite into martensite. This caused an increase in the specimen hardness due to the transformation of soft austenite into hard martensite (Figure 3). In these specimens, the dislocation density of the martensite was found to increase as a result of DCT (Figure 4). This could be due to the volume change associated with the newly transformed martensite or could be due to some other effect. This behavior can be more clearly examined on those alloys that were fully martensitic in the water-quenched condition. In these specimens, no transformation takes place during DCT. For these fully martensitic alloys, DCT was also observed to increase the dislocation density of the martensite. This is correlated with an increase in specimen hardness. We can examine if the increase in dislocation density correlates with the measured increase in hardness by plotting the hardness against the square root of dislocation density [68], as shown in Figure 13. It was seen that broadly speaking, the increase in hardness does scale with the square root of the dislocation density, indicating that the increased hardness is reasonably attributed to the increase in dislocation density.

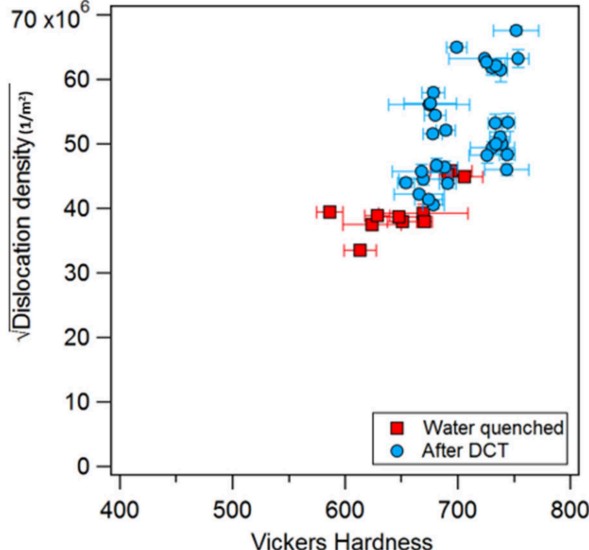

**Figure 13.** The correlation between hardness and the square root of the measured dislocation density.

The underlying reason that exposure to cryogenic temperatures increases the dislocation density in martensite is not immediately obvious, but thermal expansion could be an explanation. The atom probe tomography carried out here showed that in the as-quenched condition, the martensite contained local clusters enriched in carbon, which is consistent with the work of Morsdorf [59]. Experimental studies on bulk alloys indicate that the thermal expansion of Fe-C alloys can be significantly affected by carbon concentration [69,70], and the difference in expansion coefficients increases with decreasing temperature. Differences in local thermal expansion have been reported to induce increased localized stress, leading to dislocation multiplication [71–73]. Therefore, it is proposed that in the present case, the heterogeneous distribution of carbon within the microstructure [59] may produce pockets that have different thermal expansion properties to their surroundings, leading to local lattice strains and dislocation multiplication when cooled to cryogenic temperatures.

### 5.2. Effect of DCT on Tempering Behavior

The DFT analysis showed that the magnetic SANS signal is only sensitive to cementite particles. Since the quenched microstructures do not have cementite, the as-quenched alloys will have similar

SANS patterns before and after DCT. However, during tempering, cementite forms. This was confirmed by the APT analysis showing the formation of stoichiometric $Fe_3C$. Any differences in the size and volume fraction of the cementite that formed during tempering will be evident in the SANS patterns. The clearly visible differences in the SANS patterns of the tempered microstructures between the quenched and DCT treated steels confirms that DCT modifies the cementite precipitation during tempering. The quantification of this observation, as shown in Figure 12, indicates that some alloys showed an increase in the fine particles, while others showed an increase in the medium-sized population.

The change in the cementite size distribution that develops during tempering may be the result of the increased dislocation density generated during DCT. Dislocations act as nucleation sites for new cementite particles, and dislocations are also fast diffusion carriers. These two effects may explain the differences observed in the particle size distribution. In those alloys with a higher volume fraction of particles after DCT, this can be explained by the faster diffusivity allowing faster precipitation kinetics. For those alloys where there was a reduction in precipitate size, this can be attributed to the increased nucleation density along the dislocation network.

We can quantify the effect of dislocation density on diffusivity [74–77], with the lattice and pipe diffusion coefficients being calculated by following the work of da Silva et al. [78] for cases where carbon is diffusing and using the work of Huang et al. [79] for the self diffusion of iron. To obtain the pipe diffusion coefficient ($D'$), the assumption was made that the activation energy for pipe diffusion was half the nominal value for the bulk [80]. Equation (5) [77] was used to calculate the effective diffusion coefficients ($D_{(Eff)}$) for the given dislocation densities ($\rho$) of this work, as shown in Figure 4. The dislocation pipe radius ($a$) was taken as the Burgers vector described in Equation (4). The effective diffusion coefficient results are shown as a percentage increase from the as-quenched condition, as shown in Table 3. To note, the results shown in Table 3 are equivalent whether the calculation is for carbon or iron diffusing through the matrix. This is because the data is shown as a percentage increase from the untreated condition, not as absolute values, and the relationship between $D_{(Eff)}$ and $\rho$ is linear when all other factors are constant.

$$D_{(Eff)} = D \left[1 + \rho \pi a^2 (D'/D - 1)\right] \tag{5}$$

**Table 3.** Percentage Increase in Effective Diffusion Coefficient as Compared to the "As Quenched" Condition. 4 Deep Cryogenic Treatment (DCT) = 4 h of Deep Cryogenic Treatment, 24 DCT = 24 h of Deep Cryogenic Treatment, 36 DCT = 36 h of Deep Cryogenic Treatment.

| Alloy | Increase in Diffusion Coefficient (%) | | |
| --- | --- | --- | --- |
| | 4 DCT | 24 DCT | 36 DCT |
| Fe-0.6C | 16.0 | 29.3 | 40.6 |
| 2Ni | 19.4 | 19.2 | 35.8 |
| 4Ni | 86.4 | 84.5 | 88.5 |
| 4Cr | 58.8 | 51.3 | 37.9 |
| 7Cr | 53.0 | 55.0 | 37.1 |
| 11Cr | 27.2 | 34.3 | 24.0 |
| D2 tool steel (980 °C) | 46.4 | 58.8 | 52.7 |
| D2 tool steel (1020 °C) | 119.0 | 106.1 | 84.5 |
| D2 tool steel (1040 °C) | 124.4 | 111.6 | 81.5 |
| D2 tool steel (1060 °C) | 177.3 | 193.7 | 172.8 |
| D2 tool steel (1080 °C) | 164.4 | 226.6 | 202.8 |

As can be seen from Table 3, the increased dislocation density that results from DCT leads to an increase in the diffusion coefficient of carbon in iron; in some cases, the rate of diffusion is doubled. This means that the diffusivity of carbon in iron is enhanced by DCT, as a result of the higher dislocation densities developed by the cryogenic treatment. Since the growth of cementite is governed by carbon

diffusion, it is proposed here that the change in effective diffusion coefficient resulting from DCT may be the reason for the modified particle size distributions that are observed after the tempering of steels subject to DCT.

## 6. Conclusions

Two groups of martensitic alloys were examined for changes induced by deep cryogenic treatment (DCT). The first group was a range of binary and ternary compositions that all contained 0.6% carbon. The second group was comprised of AISI D2 tool steel specimens. The main conclusions from examining the effect of DCT on steel microstructure are as follows:

- In those alloys containing retained austenite, DCT transformed a substantial quantity of austenite to martensite.
- DCT resulted in an increase in alloy hardness due to two factors: an increase in martensite volume fraction and an increase in martensite dislocation density.
- It was found that the dislocation density of the martensite increased in all specimens, even those that did not exhibit transformation during DCT. This is believed to be the first such observation.
- The increase in dislocation density of the martensite that results from DCT is proposed to be due to local differences in thermal expansion within the complex martensite structure.

All alloys were subject to a tempering heat treatment. The following was observed:

- Tempering was shown to change the SANS scattering signal in all alloys. Quantification of the SANS signal confirmed that the size and volume fraction of cementite in all alloys was measurably changed as a result of DCT prior to tempering.
- It is suggested that the change in dislocation density that results from DCT causes the change in cementite size distribution. Calculations showed that in some cases, the diffusivity of carbon was doubled by the increase in dislocation density, and this change in the diffusivity of carbon may be responsible for the different nucleation and growth behavior of cementite.

**Author Contributions:** Conceptualization, N.E.S., D.F. and A.A.; methodology, N.E.S. and A.A., software, R.M.; validation, N.E.S., D.F., A.A.; formal analysis, A.A., A.S., R.M., N.M.S.; investigation, A.A., A.S., R.M.; resources, N.E.S., D.F.; data curation, A.S., R.M., N.E.S.; writing—original draft preparation, A.A., N.M.S.; writing—review and editing, N.M.S., N.E.S., D.F.; visualization, A.A., N.M.S., A.S., R.M.; supervision, N.E.S., D.F.; project administration, N.E.S., D.F.; funding acquisition, N.E.S., D.F. All authors have read and agreed to the published version of the manuscript.

**Funding:** This research received no external funding.

**Acknowledgments:** The authors would like to thank the University of South Australia and Deakin University for their support in writing this paper. The authors gratefully acknowledge the granting of neutron beamtime from ACNS, ANSTO (Proposal Number P4500), and the first principles calculations were supported by computational resources provided by the Australian Government through NCI under the National Computational Merit Allocation Scheme (NCMAS, Project mk63). The continued support of Prof Emily Hilder and Prof Peter Hodgson is gratefully acknowledged.

**Conflicts of Interest:** The authors declare no conflict of interest.

## Appendix A

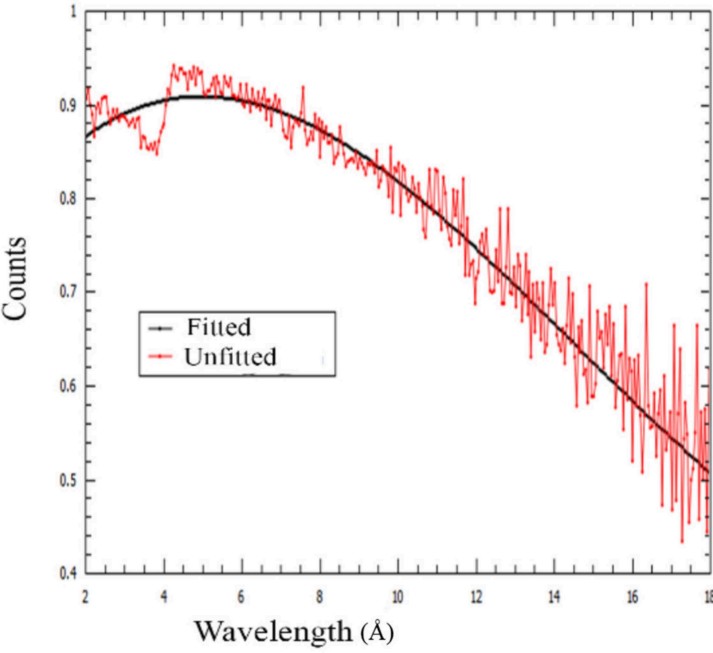

**Figure A1.** Transmission data from an as-quenched Fe-0.6C binary alloy. A magnetic field of 1.0 T was applied horizontally. The data show Bragg's edge at 3.5 Å and multiple scattering above 10 Å.

## Appendix B

Very strong scattering materials, such as steel, are prone to show a multiple scattering effect, i.e., a multiple scattering of one neutron inside the sample. Such neutrons are corrupting the scattering data. It is known that neutrons with longer wavelengths are more prone to be scattered more than once inside the sample. The Bilby instrument is utilizing a wide spectrum of wavelengths, and a careful check of the data is needed to be sure only non-corrupted data are taken for analysis. To check whether the multiple scattering is presented in the data, we are estimating the number of scattered neutrons comparing to transmitted neutrons and also analyzing the I(Q) data of reduced wavelength slices. Details of the procedure are described in [49], Section 4.3. The figure below presents an example of contaminated data. Here, the I(Q) is a scattering intensity for pure iron that is reduced on different wavelengths. One can see that only data on wavelengths below 10 Å are overlapping well and hence can be considered for further analysis.

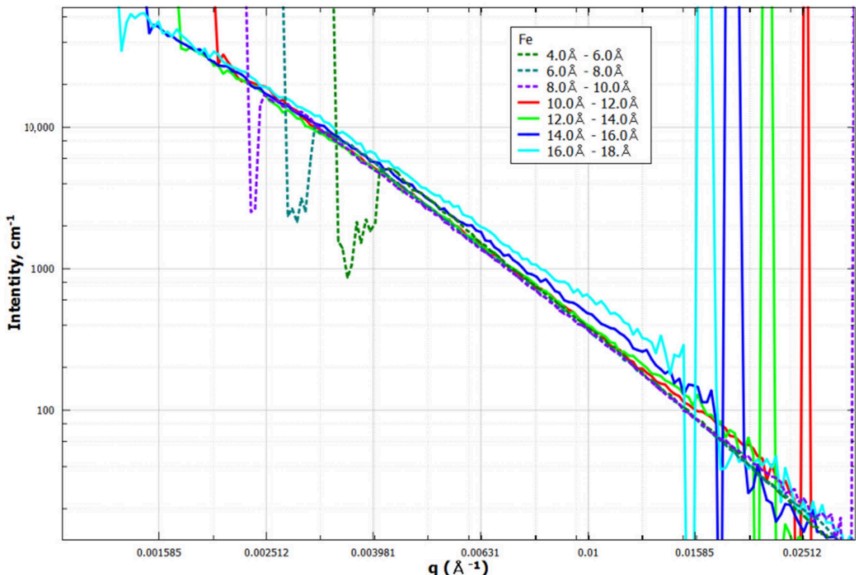

**Figure A2.** Typical example of contaminated neutron scattering data.

## Appendix C

The interaction magnetic potential of neutron with the magnetic moment of $\mu_N$ and an atom with the induced field of $B(r)$ at the neutron position $r$ can be expressed as $V(r) = -\mu_N.B(r)$ [81]. The Fourier transform of the interaction potential is called the *magnetic scattering* denoted by $b_M$ [56].

The induced magnetic field $B(r)$ can be given as the sum of the field induced by the electrons magnetic dipole moment and that induced by the electrons orbital moment denoted by $B_s(r)$ and $B_L(r)$, respectively. In the presence of a static external field $H$, the total local magnetization at point $r$ is expressed as $M(r)$ and the total local induced field can be given as $B(r) = \mu_0(H(r) + M(r))$.

The Fourier transform of the total local magnetization is $B(q) = \mu_0 M(q) sin(\theta)$ where $\theta$ is the angle between $M$ and the scattering vector $q$. It can be shown that $b_M \propto M(q)$ [56], and since the latter is the Fourier transform of $M(r)$, and $M(r) = \chi H(r)$, where $\chi$ is the magnetic susceptibility of the material under examination, $b_M \propto \chi H(q)$. As the static external field $H(q)$ is constant, $b_M \propto \chi$, i.e., the magnetic scattering that is measured in the neutron scattering experiment is proportional to the magnetic susceptibility of the material being studied. Magnetic susceptibility can take a vast range of values, e.g., from $2 \times 10^5$ for iron [57] to $2.2 \times 10^{-5}$ for aluminum and $-1.4 \times 10^{-5}$ for graphite [82].

Since the magnetic scattering signal is directly proportional to the magnetic susceptibility of the material under study, the scattering cross-section of the phases in steel can be ranked $\chi(\text{Fe}) > \chi(\text{Fe-C}) > \chi(perlite) > \chi(Fe_3C)$.

## Appendix D

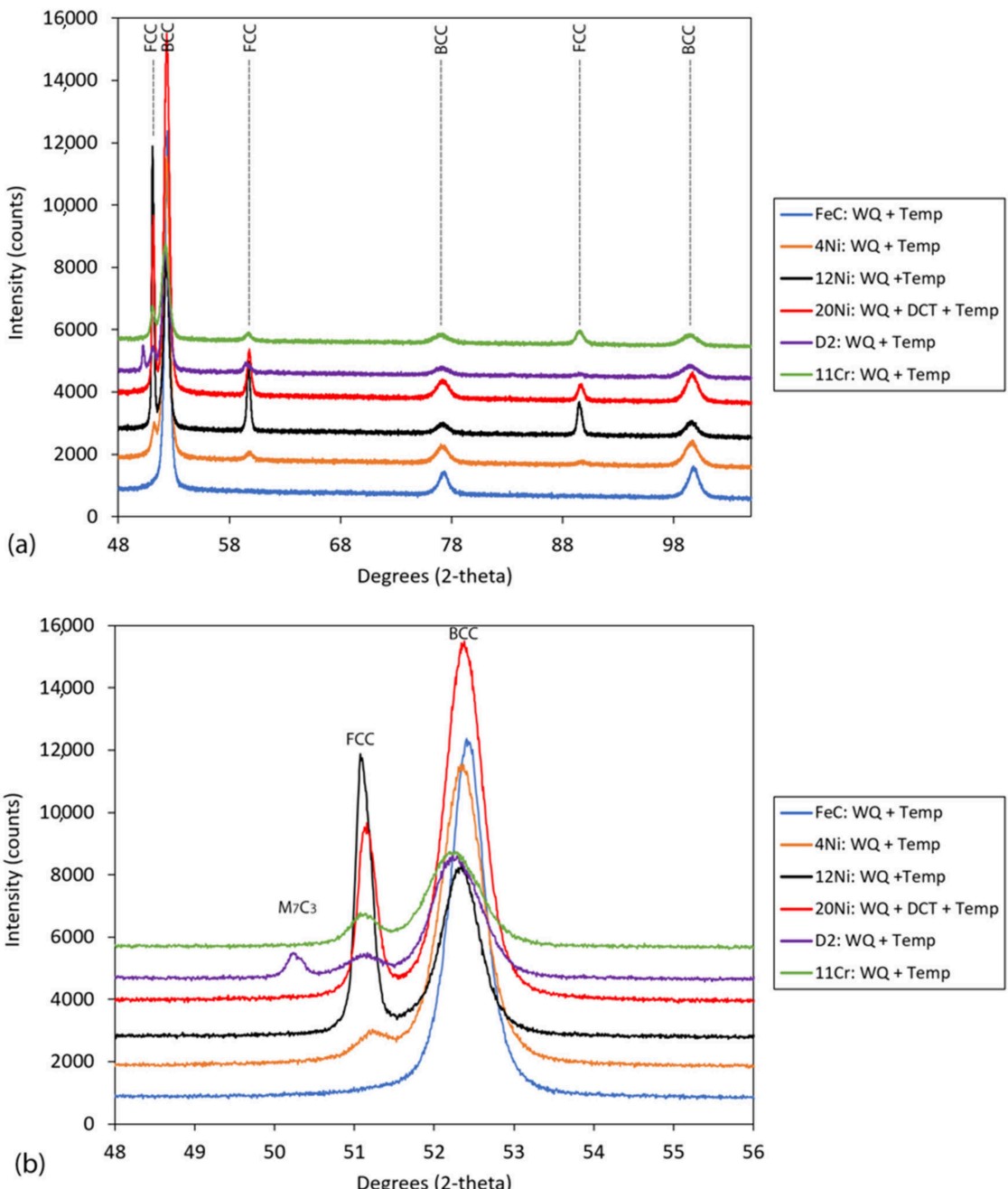

**Figure A3.** X-ray diffractograms of selected specimens as indicated in the figure legend. Note the lack of austenite in the binary Fe-0.6C alloy. Data from (**a**) are re-plotted in (**b**) showing the peaks of highest intensity in more detail. Note that these diffractograms were measured using Cobalt Kα radiation.

## Appendix E

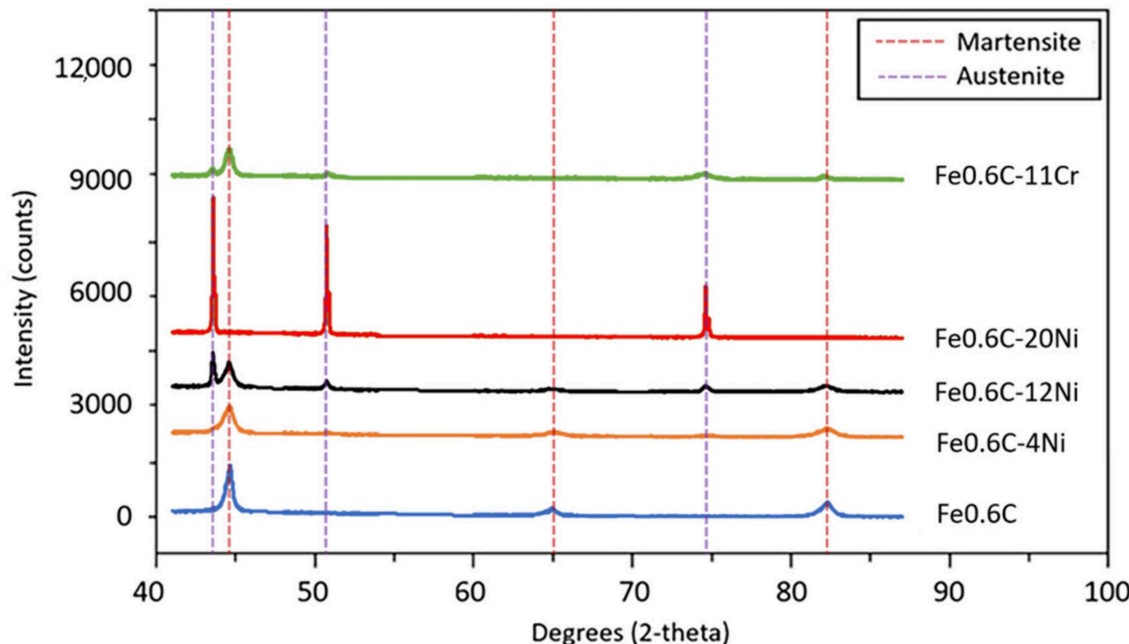

**Figure A4.** XRD diffractograms of selected alloys in as-quenched condition. Martensite and austenite peaks are indexed in the figure.

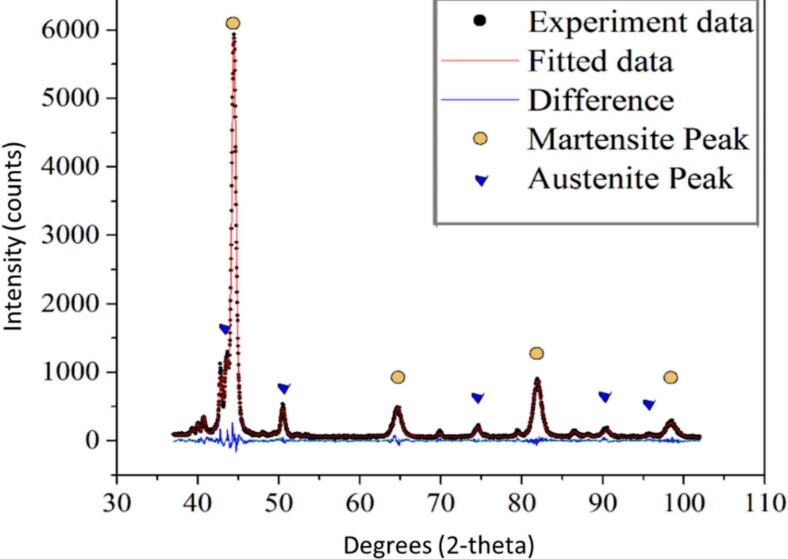

**Figure A5.** XRD diffractogram of AISI D2 tool steel austenitized at 1020 °C in as-quenched condition. The fitted model (red line) obtained from TOPAS is in good acceptance (blue line) with the experimental data (black dots). Martensite and austenite are indexed in the figure.

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
