# Peer review of "Quantification of the Dislocation Density, Size, and Volume Fraction of Precipitates in Deep Cryogenically Treated Martensitic Steels"

_metals, doi:10.3390/met10111561_

Round 1

Reviewer 1 Report

Good paper I agree with your conclusion

Line 214-215: Are these parameters values from reference 67 ?, if yes, please clarify

Author Response

We agree with the reviewer that this should be clarified, and line 211 now reads:

using the following relationship and values for the burgers vector and k [67]

Reviewer 2 Report

A high quality scientific paper. The introduction is clear. The methodology is properly selected; advanced techniques including atom probe tomography are applied. A wide range of laboratory produced alloys (various contents of Cr and Ni) and an industrial alloy was examined. Results provide both experimental and computational evidences on a type and distribution of identified carbides. The results are interestingly discussed in reference to literature studies.

Minor remarks:

1/ Title.

Actually, particle size distribution using small angle neutron scattering (SANS) and atom probe tomography was also examined after tempering, not only after a cryogenic cooling. Consider, if the title can be changed including … tempering…, in a tempered state.

2/ Figure 13; Should be a unit in the vertical axis in this figure ? Dislocation density.

Author Response

1) We have chosen not to include tempering in the title as the focus of the paper is on the DCT rather than tempering step, and only some of the data was presented post tempering. However, as it is still a significant part of the paper the word ‘tempering’ is one of the keywords for this article and tempering is also discussed in the abstract.

2) The vertical axis in the figure has now been added.

Reviewer 3 Report

Dear editor,

The authors have addressed all my points in the cover letter. Very little has changed in relation to my concerns on the applied methods. But that is a question of opinion then, and the manuscript contains all details that are necessary to the readers to judge the work him/her-selves.

Moreover, I think that the presented conclusions are justified independent on the details of the applied methods applied.

Consequently, I think that the manuscript can be published in the present form.

Regards

P.S. (Few remarks to the authors):

1. “Quantitative phases analysis was carried out using TOPAS 5.0 using the full diffractogram.”

Ok using a program as a black box tool, but it would not be a bad idea to check what the program actually does!

2. "The contribution to peak broadening of any crystal above 500 nm in diameter is negligible, therefore we chose to assume all broadening is due to dislocation density."

In commercial steels, the martensite features can be much smaller than that, not surprisingly as small as a few dozen nm.

3. "As stated in the manuscript – APT was carried out to search for chromium-carbides, and none were observed, We believe this is due to the low tempering temperature, usually secondary hardening is associated with higher tempering temperatures, generally in the order of 400ºC."

Fe transition carbides do not contain Cr! Rather their chemistry is Fe2.4-C, which is not so far from the one of cementite. That is the reason why diffraction techniques (TEM, S-XRD) are used. Alternatively, hardness measurements can give a hint on the presence of such small precipitates.

4. The work of D. Das has been shown flawed in more than one publication.

Author Response

We thank the reviewer for their insights and comments.

This manuscript is a resubmission of an earlier submission. The following is a list of the peer review reports and author responses from that submission.

Round 1

Reviewer 1 Report

Good paper.

Line 205-208: it is suggested to add the dimension of all parameters in the explanation of relationship 4, not only that of Burger vector.

Reviewer 2 Report

Very interesting work with high significance for all researchers working on cryogenic treatment of steel grades

Reviewer 3 Report

Dear editor and authors,

The manuscript entitled “Quantification of the dislocation density, size and volume fraction of precipitates in deep cryogenically treated martensitic steels” is an original and very interesting piece of work. However, I have major reserves on the applied methodology and, consequently, on the presented experimental results. In the following I have elaborated my concerns.

Section 2: material and methods

  • It is well established that martensite can form at cryogenic temperature both during cooling, isothermal holding and heating (look-up the work by M. Villa and M.A.J. Somers, Technical University of Denmark). Details about the heating rates from -196C should be added to fully evaluate the effect of the applied cryogenic treatment;

  • Small angle neuron scattering:

  • D2 steel contains a large fraction of M7C3 carbides that are most likely non-magnetic. How do these carbides contribute to the signal?
  • And what about the signal from epsilon/eta Fe transition carbides that are instead ferromagnetic. Do you expect to contribute the signal similarly to cementite?

  • XRD:
    • Information on which reflections of austenite and martensite were used to evaluate the fraction of retained austenite should be included. Moreover, the authors should include some details on the quantification method: which R factors were used? While standard values for Cr and Mo radiations can easily found, I´m not aware of standard values for Co radiation.
    • Information should be added on which reflection was used to evaluate the strain broadening.
    • Micro-strain where quantified within the assumption of negligible contribution of the crystalline size to line broadening. While this is most likely the case for coarse grain materials, like I expect all the model alloys, this approximation appear very unlikely to hold for D2. Also, very small martensite crystals may be present in the Fe-0.6C steel. In my opinion, the authors should have considered the Williamson-Hall method corrected with the use of dislocation contrast factors to evaluate the micro-strain (look-up the work by T. Ungar´s group, Eotvos University).
    • Information should be added on how the authors handled the tetragonality of martensite in relation to peak fitting. How was the doublet of martensite de-convoluted to evaluate peak broadening?

Section 3: results

  • Retained austenite:
    • It is very well established that a complete elimination of the fraction of retained austenite is impossible in medium-high carbon steel (essentially all the steels with a C content higher than 0.3wt%). The authors claim several times that a fully martensitic structure is obtained in the steel with the lowest content of alloying elements, essentially Fe-0.6C, after quenching to room temperature. This claim should be smoothen: the content of retained austenite could be defined, at the very end, as minor (2%?) for this steel.
    • In connection to the above reported point, the authors present a series of diffractograms in appendix D. These refer to the materials after tempering (as readily visible from the absence of the martensite doublets for the b.c.c. structure). Here no austenite could be detected for Fe-0.6C. This is not surprising because tempering at 200C promotes a decomposition of retained austenite in C steel (look-up the work by, for example, E.J. Mittemeijer´s group from the late 1980ies, or the one from Speich and Krauss in the 1970-1980ies) while this is not the case in alloyed steels (for example D2 requires a tempering at temperatures higher than 500C to obtain the decomposition of retained austenite). The author should present also the diffractograms acquired before tempering to support their claims.
    • Error bars should be reported throughout the all manuscript and considerations regarding the experimental error should be added in the text. How much the fitting error is relevant v.s. the variations that could be observes among identically treated samples / various location in the same sample.

Section 4: tempering

  • Tempering:
    • The stages of tempering of steel are very well established, but the authors seems not to be aware of this knowledge. In C steels: stage 0, -40C to 120C, consists in C clustering; stage I, in the precipitation of epsilon/eta Fe transition carbides, takes place in the interval 120-200C; stage II and III, decomposition of retained austenite and precipitation of cementite, respectively, are observed at temperatures between 200C and 300C. In alloyed steels, stages 0, I and III may be suppressed, while stage II may occur at a temperature significantly higher (500-550C). Stage IV can also be observed upon tempering at temperatures higher than 500C. The authors are invited to re-evaluate their results based on the above reported input. What about transition carbides? These have always been central in the discussion on the effect of cryogenic treatments.
    • SANS data indicate a bi-modal size distribution of the carbide particles in the steels. The dimension of such particles appear very consistent with expectations on the dimension of epsilon/eta Fe transition carbides (the smallest) and cementite (the largest), respectively. Can the authors elaborate on this?
    • A vary fast assessment by SEM can easily prove the presence of a bimodal particle distribution. The author should consider whether this type of investigation could be an asset to include in the manuscript. I think it would because it would be very convincing, on top of the elaborated interpretation of SANS signal.
  • Retained austenite:
    • In section 4.2.2.: “four alloys had composition such that they were almost fully martensticis in the as-quenched condition, thus cryogenic treatment had minimal changes on the martensite fraction. In these cases, FeC, 4Ni, 11Cr, D2, …”. First in Fe-4Ni-0.6C the presence of retained austenite is significant (Fig. in appendix D); second, for Fe-0.6C-11%Cr the material is expected to be almost fully austenitic upon quenching, unless the heat treatment was incorrectly performed; third, for D2 the authors mention in section 2 a minimum content of retained austenite of 10%, which in my opinion is not negligible. The author should re-evaluate this claim.

Section 5: discussion

  • c.c. metals are very brittle at cryogenic temperatures. The authors indicate the possibility that localized thermal stresses could generate lattice strain (i.e. plastic phenomena). Why should plasticity take over in brittle materials? Wouldn´t the formation of micr-cracks be more likely?
  • he formation of cementite is considered rate-controlled by the Fe diffusion (based on the work by E.J. Mittemeijer`s group). Can the author support with literature data that C diffusion is rate controlling, instead? Alternatively Fe diffusion, rather than C diffusion should be considered.